# Pleiotropic Roles of a KEAP1-Associated Deubiquitinase, OTUD1

**DOI:** 10.3390/antiox12020350

**Published:** 2023-02-01

**Authors:** Daisuke Oikawa, Kouhei Shimizu, Fuminori Tokunaga

**Affiliations:** Department of Medical Biochemistry, Graduate School of Medicine, Osaka Metropolitan University, Osaka 545-8585, Japan

**Keywords:** antiviral response, cancer, cell death, deubiquitinating enzyme, KEAP1, OTUD1, ubiquitin

## Abstract

Protein ubiquitination, which is catalyzed by ubiquitin-activating enzymes, ubiquitin-conjugating enzymes, and ubiquitin ligases, is a crucial post-translational modification to regulate numerous cellular functions in a spatio–temporal-specific manner. The human genome encodes ~100 deubiquitinating enzymes (DUBs), which antagonistically regulate the ubiquitin system. OTUD1, an ovarian tumor protease (OTU) family DUB, has an N-terminal-disordered alanine-, proline-, glycine-rich region (APGR), a catalytic OTU domain, and a ubiquitin-interacting motif (UIM). OTUD1 preferentially hydrolyzes lysine-63-linked ubiquitin chains in vitro; however, recent studies indicate that OTUD1 cleaves various ubiquitin linkages, and is involved in the regulation of multiple cellular functions. Thus, OTUD1 predominantly functions as a tumor suppressor by targeting p53, SMAD7, PTEN, AKT, IREB2, YAP, MCL1, and AIF. Furthermore, OTUD1 regulates antiviral signaling, innate and acquired immune responses, and cell death pathways. Similar to Nrf2, OTUD1 contains a KEAP1-binding ETGE motif in its APGR and regulates the reactive oxygen species (ROS)-mediated oxidative stress response and cell death. Importantly, in addition to its association with various cancers, including multiple myeloma, OTUD1 is involved in acute graft-versus-host disease and autoimmune diseases such as systemic lupus erythematosus, rheumatoid arthritis, and ulcerative colitis. Thus, OTUD1 is an important DUB as a therapeutic target for a variety of diseases.

## 1. Introduction

Ubiquitination, which is catalyzed by three kinds of enzymes, ubiquitin-activating enzyme (E1), ubiquitin-conjugating enzyme (E2), and ubiquitin ligase (E3), is one of the major post-translational modifications. The human genome encodes two E1s, ~40 E2s, and >600 E3s [1]. Combinations of these enzymes generate monoubiquitination and/or multiple types of polyubiquitin linkages via seven internal lysine (Lys, K) residues, K6, K11, K27, K29, K33, K48, and K63, and the N-terminal methionine-1 (Met1, M1). In addition to homotypic polyubiquitin chains, those with complex structures such as branched and hybrid chains account for a large portion (10–20%) of the ubiquitin in polymers [2]. Furthermore, post-translational modifications of ubiquitin itself, including phosphorylation, acetylation, and ADP-ribosylation, contribute to the regulation of the ubiquitin system. The variety of ubiquitin architectures and modifications, collectively referred to as the “ubiquitin code”, regulates numerous cellular functions such as proteasomal degradation, signal transduction, DNA repair, membrane trafficking, and epigenetics [1]. Furthermore, in addition to protein ubiquitination at the ε-NH_2_-group of internal Lys residues through an isopeptide bond, non-Lys ubiquitinations such as the thioester-linked ubiquitination of cysteine (Cys) residues and oxyester-linked ubiquitination of serine (Ser)/threonine (Thr) residues, and the non-protein ubiquitinations of lipopolysaccharide (LPS), glycogen, ADP-ribose, and phosphatidylethanolamine, have been identified [3,4,5]. Studies on the diversity of ubiquitination are expected to provide further interesting features.

Deubiquitinating enzymes (DUBs) reversibly regulate the ubiquitin system as “erasers” of the ubiquitin code. There are about 100 human DUBs, which are classified into seven subfamilies: 56 members of ubiquitin-specific proteases (USPs), 17 ovarian tumor proteases (OTUs), 4 ubiquitin C-terminal hydrolases (UCHs), 4 Machado–Josephin domain-containing proteases (MJDs), 5 motif interacting with ubiquitin-containing novel DUBs (MINDYs) [6], 1 zinc finger with the UFM1-specific peptidase domain protein (ZUFSP) [7], and 12 JAB1/MPN/Mov34 metalloenzymes (JAMM/MPN+) [8,9]. The USP, OTU, UCH, MJD, MINDY, and ZFUBP proteins are cysteine proteases, whereas the JAMM/MPN+ family proteins are zinc metalloproteases. DUBs have different ubiquitin linkage specificities, catalytic activities, and subcellular localizations. Therefore, DUBs are important in the spatiotemporal regulation of cellular functions, and failures of the DUB system are associated with diseases [1,9,10].

In this review, we comprehensively summarize the information available for OTUD1, an OTU family DUB, which participates in the regulation of cancer progression, antiviral signaling, innate and acquired immune responses, reactive oxygen species (ROS)-mediated oxidative damage through kelch-like ECH-associated protein 1 (KEAP1)-binding, and multiple related functions by cleaving diverse ubiquitin linkages. Therefore, OTUD1 dysfunction is involved in various diseases.

## 2. OTU Family DUBs

The homologs of the ovarian tumor (*otu*) gene product in *Drosophila melanogaster*, which is required for oocyte morphogenesis, were identified as a cysteine protease superfamily in eukaryotes, several groups of viruses, and the pathogenic bacterium, *Chlamydia pneumoniae* [11,12]. The human genome has 17 OTU family DUBs, which are further divided into four subfamilies: the otubain subfamily (OTUB1 and OTUB2), the OTUD subfamily (OTUD1/DUBA7, OTUD2/YOD1/DUBA8, OTUD3/DUBA4, OTUD4/DUBA6, OTUD5/DUBA, OTUD6A/DUBA2, OTUD6B/DUBA5, and ALG13), the A20-like OTU subfamily (A20/TNFAIP3, OTUD7A/Cezanne2, OTUD7B/Cezanne, TRABID/ZRANB1, and VCPIP1/DUBA3/VCPIP135), and the OTULIN subfamily (OTULIN/FAM105B and FAM105A/OTULINL) (Figure 1) [13,14].

Some OTU family DUBs exhibit stringent ubiquitin linkage specificities in vitro, such as OTUD7A and OTUD7B to K11, OTUD4 and OTUB1 to K48, OTUD1 to K63, and OTULIN to M1 [13]. Among the OTU family DUBs, TRABID reportedly has an esterase activity to cleave Ub~Thr [15]. FAM105A/OTULINL is a structurally inactive pseudodeubiquitinase [16], but seems to have weak DUB activity toward some ubiquitin linkages [17]. Several OTU family DUBs have ubiquitin interaction domains, such as ubiquitin-interacting motif (UIM), ubiquitin regulatory X (UBX), ubiquitin-associated (UBA), zinc finger (ZF), and ankyrin repeat ubiquitin-binding domain (AnkUBD) (Figure 1). OTU family DUBs regulate cell signaling cascades and are involved in human diseases, including cancer, inflammation, neurodegenerative diseases, and viral infections. For example, high mRNA expression levels of *OTUD1, OTUD3, OTUD4*, and *ALG13* are associated with improved prognosis in non-small cell lung cancer (NSCLC) and adenocarcinoma, but not in squamous cell carcinoma [18]. By contrast, high expression levels of *OTUD2* mRNA are associated with poor survival in NSCLC patients. In this review, we will focus on OTUD1.

## 3. Pleiotropic Cellular Functions of OTUD1

Human *OTUD1* (also known as DUBA7 and OTDC1) is encoded on chromosome 10p12.2 as a single exon, and its gene product, OTUD1, is composed of 481 residues with a molecular mass of 51 kDa. The N-terminal region of OTUD1 (aa 1–290) contains an abundance of alanine (Ala), proline (Pro), and glycine (Gly), and, therefore, we named it the Ala-, Pro-, and Gly-rich region (APGR) [19]. APGR is an intrinsically disordered, low-complexity domain. OTUD1 also includes a catalytic OTU domain (aa 291–446) with the active Cys320 and a UIM domain (aa 457–481) at the C-terminus (Figure 1). Importantly, full-length OTUD1 and a construct of OTU + UIM are highly active and K63-linked ubiquitin chain-specific in vitro, whereas a construct of the OTU domain alone shows reduced activity and specificity, indicating that UIM is important for the specificity and efficiency of OTUD1 towards the K63-linkage [13]. The recombinant OTUD1 synthesized by a wheat germ cell-free system shows strong affinity for the K63-ubiquitin chain but also hydrolyzed K48- and K6-linked ubiquitin chains in vitro [17]. As described later, cellular analyses suggested that OTUD1 is involved in the cleavage of various ubiquitin linkages.

### 3.1. OTUD1 Functions as a Tumor Regulator

Several reports linked OTUD1 to cancers. OTUD1 was initially characterized as a biomarker for thyroid carcinoma by a single-chain variable fragment antibody-reactive antigen, using two-dimensional polyacrylamide gel electrophoresis followed by mass spectrometry, although a detailed functional analysis of OTDU1 in thyroid cancer has not been performed [20]. OTUD1 reportedly binds and stabilizes p53, a crucial tumor suppressor, in a DUB-activity-dependent manner [21]. In this process, OTUD1 preferentially removes the K48-linked ubiquitin chain over the K63 chain, and the OTUD1-mediated cell cycle arrest then leads to apoptosis (Figure 2A). These results suggest that OTUD1 is a novel regulator of p53 and its aberrant function and/or decreased expression destabilize p53, leading to oncogenesis. Zhang et al. reported that the shRNA-mediated knockdown of *OTUD1* in MDA-MB-231 human breast cancer cells leads to the production of strong metastatic nodules in xenografted mice, indicating that OTUD1 is a metastasis-repressing factor [22]. On a cellular basis, OTUD1 inhibits the transforming growth factor-β (TGF-β)/small mothers against decapentaplegic (SMAD) signaling pathway and epithelial-to-mesenchymal transition (EMT). OTUD1 stabilizes SMAD7, a TGF-β pathway inhibitor, by removing the RNF12-mediated K48-linked ubiquitin chain. Moreover, OTUD1 reinforces SMAD7–SMURF2 complex formation by removing the K33 ubiquitin chain at K220 near the PY motif, the SMURF2-binding site of SMAD7, and is involved in the turnover of TGF-β receptor I from the cell surface (Figure 2B). The authors also show that the loss of *OTUD1* correlates with the poor prognosis of many tumor types, including glioblastoma, melanoma, and lung and breast cancers. Thus, OTUD1 is a crucial metastasis suppressor in various tumors.

Downregulation of OTUD1 is also involved in the poor prognosis of clear cell renal cell carcinoma (ccRCC) [23]. Through their database analysis, Liu et al. note that among the OTU family DUBs, the mRNA levels of *OTUD1, OTUD2, OTUD6B*, and *OUTD7B* are downregulated in ccRCC, while those of *OTUD3, OTUD6A*, and *A20* are upregulated, and the downregulation of *OTUD1* is the key for the poor prognosis of ccRCC patients. OTUD1 negatively regulates the cell cycle to inhibit the abnormal proliferation of cancer cells, since OTUD1 interacts with phosphatase and tensin homolog (PTEN), a negative regulator of the AKT (also known as protein kinase B) and NF-κB signaling pathways, and regulates its stability (Figure 2C). PTEN affects the sensitivity of cancer cells to anticancer drugs, such as AKT inhibitors, MEK inhibitors, and tyrosine kinase inhibitors. Indeed, OTUD1 participates in the sensitivity of sunitinib, a receptor tyrosine kinase inhibitor, through PTEN. Therefore, the OTUD1–PTEN axis suppresses tumor growth and regulates the resistance of ccRCC to tyrosine kinase inhibitors. Very recently, Fan et al. showed that OTUD1 binds AKT most strongly among 11 OTU family DUBs, and then suppresses its phosphorylation [24]. Importantly, the authors identified that the AKT-binding site of OTUD1 locates within the disordered APGR (aa 92–127) (Figure 2D). A 36 residue peptide (designated OUN-36) derived from this region binds to the PH domain of AKT and disturbs the interaction with phosphatidylinositol (3,4,5)-triphosphate (PIP3). OUN-36 inhibits the growth of AKT-hyperactive tumor cells. The ONU-36 fusion protein suppresses AKT signaling and tumorigenesis, indicating the availability of peptide-based cancer therapy.

OTUD1 is involved in iron metabolism and antitumor immunity. Iron-sensitive element-binding protein 2 (IREB2, also known as IRP2) is an mRNA-binding protein that regulates iron homeostasis [25]. Song et al. showed that OTUD1 binds IREB2 through the N-terminal APGR, and deubiquitinates K48- and K63-linked ubiquitin chains from IREB2. The stabilized IREB2 then induces transferrin receptor protein 1 (TFRC) production and promotes iron transport into cells, which leads to increased ROS production and ferroptosis, iron-mediated cell death [26] (Figure 2E). OTUD1 also promotes the release of damage-associated molecular patterns (DMAPs), which strengthens the host immune response. Importantly, the selective downregulation of OTUD1 in colon cancer facilitates the proteasomal degradation of IREB2, which results in reduced TFRC. The impaired iron metabolism causes ROS production and supports the progression of colorectal cancer. Thus, OTUD1 functions as a regulator of host antitumor immunity through the regulation of the OTUD1–IREB2–TFRC pathway.

**Figure 2 antioxidants-12-00350-f002:**
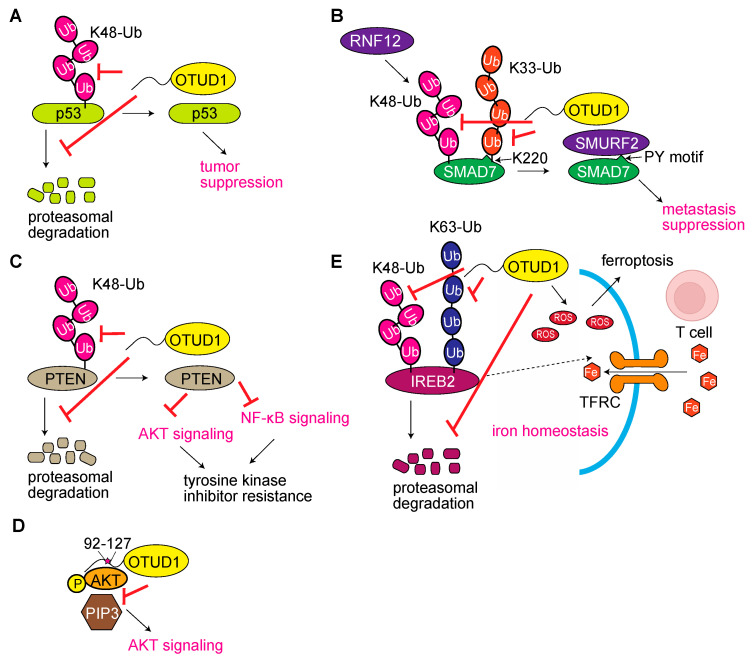
Effects of OTUD1 on tumorigenesis. (**A**) OTUD1 stabilizes p53 [21]. (**B**) Deubiquitination of SMAD7 by OTUD1 inhibits the TGF-β/SMAD signaling pathway and suppresses metastasis [22]. (**C**) OTUD1 stabilizes PTEN, resulting in the suppression of the AKT and NF-κB signaling pathways [23]. (**D**) OTUD1 binds AKT through the N-terminal aa 92–127 and inhibits its activity [24]. (**E**) OTUD1 regulates iron metabolism through the deubiquitination of IREB2 and suppresses colorectal cancer [26].

In other papers suggesting a link between OTUD1 and cancer, OTUD1 reportedly deubiquitinates and stabilizes Krüppel-like factor 4 (KLF4), a zinc finger transcription factor that regulates proliferation, differentiation, apoptosis, and reprogramming [27]. The downregulation of *OTUD1* and the destabilization of KLF4 may be involved in NSCLC progression. Thus, KLF4 seems to be a substrate of OTUD1. Although the details remain enigmatic, in a comparison of cancer exosome-transfected OK113 normal primary oral keratinocytes with untransfected cells, *OTUD1* was identified as one of the decreased genes altered by exogenous exosomes [28]. Therefore, OTUD1 may be selectively released and eliminated from the cancer cells by exomes. Moreover, a recent study based on glioma cases showed that the increased expression of OTUD1 in tumor associated-macrophages is associated with a risk of poor prognosis [29], although the detailed cellular mechanism is unknown.

### 3.2. OTUD1 Regulates Yes-Associated Protein (YAP) Signaling

YAP is an important transcription co-activator involved in the regulation of organ size, tissue homeostasis, and tumorigenesis through the Hippo pathway [30]. YAP, as well as a transcriptional co-activator with a PDZ-binding motif (TAZ), cannot bind DNA directly, but it interacts and potentiates TEAD family of transcription factors. Yao et al. showed that K63-ubiquitination at K321 and K497 in YAP by a complex type E3, SCF^Skp2^, is associated with the nuclear localization and upregulation of the mRNA levels of YAP target genes through increased interactions with TEAD (Figure 3A) [31]. OTUD1 binds to YAP and removes its K63-ubiquitin chain in a Hippo-independent manner, indicating that OTUD1 is a negative regulator of YAP signaling. Importantly, a xenograft of MDA-MB-231 cells expressing YAP with mutated Lys residues that accept K63 ubiquitination showed attenuated tumorigenesis, whereas the knockdown of *OTUD1* enhanced tumorigenesis, suggesting that OTUD1 is an important inhibitor of tumorigenesis. Grattoarola et al. reported that chemoresistant pancreatic ductal adenocarcinoma (PDAC) cell lines, such as PANC-1 cells, expressed higher levels of nuclear factor erythroid 2-related factor 2 (Nrf2), HO-1, YAP, and survive longer than those in chemosensitive PDAC cell lines with reduced oxidative stress [32]. The slowed degradation of Nrf2 and YAP with the high expression of USP17 and OTUD1 was detected in the chemoresistant PDAC cell line, and the knockdown of *OTUD1* reduced the levels of Nrf2 and YAP, and cell growth. Thus, OTUD1 seems to be involved in the malignant progression of PDAC through the stabilization of Nrf2 and YAP. In addition, Liu et al. showed that the expression level of OTUD1 is downregulated and that of SOX9, a transcription factor that plays an essential role in sexual differentiation and chondrocyte differentiation, and SPP1, which encodes osteopontin, are inversely correlated and downregulate SOX9 and SPP1 in NSCLC [33]. OTUD1 suppresses resistance to erlotinib, an epidermal growth factor receptor tyrosine kinase inhibitor, in NSCLC cells and tumor-bearing mice. OTUD1 inhibits the nuclear translocation of YAP through deubiquitination, thereby attenuating its activity. Thus, OTUD1 functions as a tumor suppressor and contributes to the efficacy of erlotinib in NSCLC through the YAP/SOX9/SPP1 axis. Together, these findings suggest that OTUD1 is a suppressor of YAP signaling and is involved in the progression and chemoresistance of cancer cells.

### 3.3. OTUD1 Regulates the Degradation of Myeloid Cell Leukemia 1 (MCL1)

MCL1 is a fast-turnover Bcl-2 family protein that plays important roles in cell survival, proliferation, differentiation, and tumorigenesis [34]. The effect of OTUD1 on MCL1 degradation is controversial. Wu et al. show that OTUD1 stabilizes MCL1 in a DUB-activity-dependent manner (Figure 3B) [35]. OTUD1 also regulates the sensitivity to a BH3-mimetic compound, ABT-263, in cancer cells, and attenuates the synergistic effect of an anticancer medicine, sorafenib. Therefore, the authors conclude that OTUD1 is a negative prognostic factor for liver, ovarian, breast, and cervical cancers. In contrast, Luo et al. screened DUBs that play essential roles in chemotherapy resistance in esophageal squamous cell carcinoma (ESCC), and identified OTUD1 as the only DUB that is significantly downregulated in chemoresistant ESCC cell lines as compared to its levels in the parental cell line [36]. OTUD1 interacts with apoptosis-inducing factor (AIF), a crucial mitochondrial protein involved in cell survival and apoptosis through NADH oxidoreductase activity. OTUD1 deubiquitinates the K27- and K63-linked ubiquitin chains on K244 of AIF and the K63-linked ubiquitin chain on K255 (Figure 3C). Interestingly, the deubiquitination of K255 promotes the nuclear translocation and DNA-binding ability of AIF, thus, resulting in caspase-independent apoptosis, referred to as parthanatos. In contrast, the deubiquitination of AIF at K244 disrupts the mitochondrial structure and compromises oxidative phosphorylation (OXPHOS). OTUD1 stabilizes DDB1 and CUL4-associated factor 10 (DCAF10), and recruits the E3 complex of CUL4A–DDB1 to facilitate MCL1 degradation and caspase-dependent apoptosis. Thus, OTUD1 seems to be a central regulator of MCL1, cell death pathways, mitochondrial OXPHOS, and chemoresistance of tumor cells. More detailed studies of the effects of OTUD1 on MCL1 are awaited.

**Figure 3 antioxidants-12-00350-f003:**
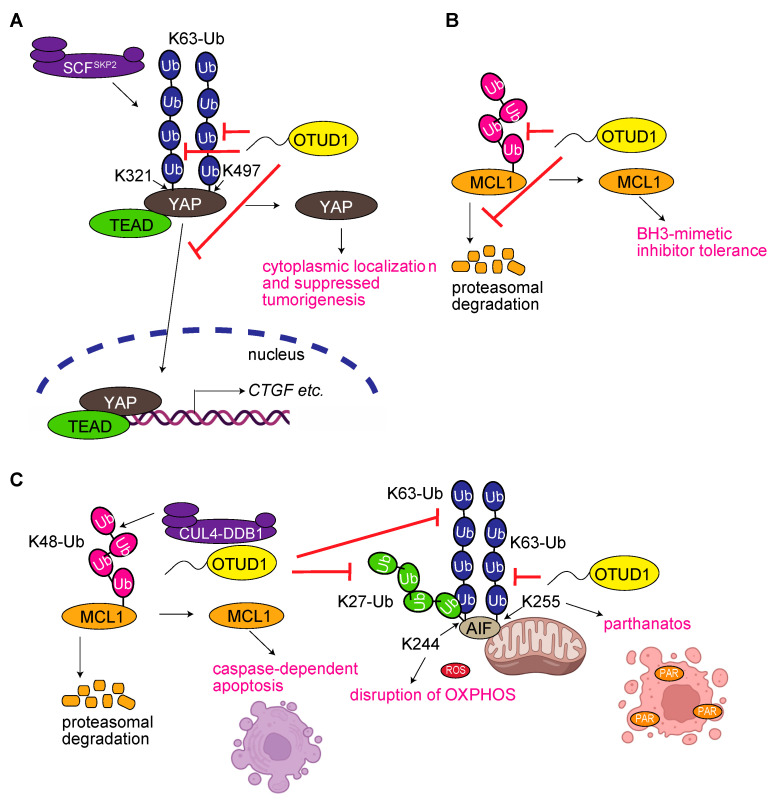
OTUD1 regulates YAP signaling and MCL1 degradation in tumorigenesis. (**A**) OTUD1 regulates the YAP signal [31]. (**B**) OTUD1 inhibits MCL1 degradation [35]. (**C**) OTUD1 regulates MCL1 degradation, mitochondrial AIF function, and cell death [36].

### 3.4. OTUD1 Regulates Antiviral and Innate Immune Responses, and Its Dysfunction Is Associated with Autoimmune and Inflammatory Diseases

OTUD1 reportedly suppresses the RNA-virus-induced production of type I interferons (IFNs) and proinflammatory factors [37]. Thus, the induction of mRNAs encoding IFNβ, TNF-α, and IL-6 was upregulated upon infection with RNA viruses such as Sendai virus (SeV) and vesicular stomatitis virus (VSV), but not by a DNA virus, herpes simplex virus (HSV), in *Otud1*-deficient mouse embryonic fibroblasts (MEFs) and mice, as compared to the responses in the wild-type (WT) counterparts. These results suggest that OTUD1 is involved in the host defense against RNA viruses. In this process, OTUD1 negatively regulates the protein levels of mitochondrial antiviral-signaling protein (MAVS) and the TNF receptor-associated factor 3 (TRAF3) and TRAF6 E3s. Furthermore, OTUD1 regulates the ubiquitination and protein level of a HECT-type E3, Smurf1, and the OTUD1–Smurf1 axis destabilizes MAVS/TRAF3/TRAF6 levels and decreases IFNβ induction during RNA virus infection (Figure 4A). In a somewhat controversial report, Zhang et al. show that OTUD1 expression is upregulated by both SeV and HSV infections, and *Otud1*-deficiency enhances the antiviral innate immune response and exhibits resistance against SeV and HSV infection [38]. OTUD1 reportedly restricts IRF3 by removing its critical K6-linked ubiquitin chain from K105 (Figure 4A). In contrast to viral infection, *Otud1^−/−^*-mice were more susceptible to LPS-induced sepsis through the negative regulation of LPS-triggered signaling, by targeting IRF3. Thus, OTUD1 seems to be a negative antiviral innate immune response regulator. Wang et al. analyzed the effect of DNA-damage-inducible transcript 3 (DDIT3, also known as CHOP), a regulator of endoplasmic reticulum (ER) stress-induced apoptosis and autophagy, on the antiviral immune response during an infection with bovine viral diarrhea virus (BVDV), a single-stranded RNA virus [39]. The authors show that BVDV infection upregulates DDIT3 expression and promotes BVDV replication and MAVS degradation. After the BVDV infection, the expression of OTUD1 is upregulated in DDIT3-overexpressing cells and inhibits IFN antiviral responses. OTUD1 negatively regulates MAVS protein levels by the upregulation of Smurf1 (Figure 4A). Therefore, the authors concluded that the DDIT3–OTUD1–MAVS pathway suppresses IFN production through the degradation of MAVS.

Patients with major depressive disorder (MDD) have a higher risk of viral infection. Zhang et al. show that depression reduces Abelson helper integration site 1 (AHI1), a critical protein for brain development in the hypothalamus and amygdala and attenuates the antiviral response [40]. Indeed, the expression of IFN-stimulated genes (ISGs) is downregulated in *Ahi1^−/−^* cells due to weakened IFN signaling and cellular antiviral immunity. Concomitant with the reduction in AHI1, Tyk2, a JAK family tyrosine kinase, is also decreased. The authors report that AHI1 binds OTUD1, which then removes the K48-linked polyubiquitin chain from Tyk2 to escape from proteasomal degradation. Depression elevates arginine vasopressin (AVP), which decreases AHI1 and Tyk2, thus, inhibiting IFN signaling and eventually attenuating the host’s antiviral innate immunity. Meptazinol, a partial μ-opioid receptor agonist, upregulates Tyk2 levels by promoting AHI1 expression, which involves cellular IFN signaling and antiviral activity in vitro and in vivo. Therefore, OTUD1 protects against virus infection in the nervous system.

Dysregulation of innate immunity by excessive IFN production is associated with autoimmune diseases. Notably, the OTUD1 mutations P95R, R243C, P379S, T407P, P420L, and G430V were identified in systemic lupus erythematosus (SLE), rheumatoid arthritis (RA), and ulcerative colitis (UC) patients (Figure 4B) [41]. Although WT OTUD1 drastically suppressed poly(I:C)-induced NF-κB and IFN-sensitive response element (ISRE) activation, the autoimmune disease-associated P95R, R243C, P379S, and P420L mutants, as well as a DUB-inactive C320S mutant, attenuated the inhibitory effects of OTUD1 on the innate immune response. The N-terminal region of OTUD1 interacts with the Pro-rich domain of IRF3. OTUD1 then removes the K63-linked ubiquitin chain from K98 of IRF3 and attenuates the RIG-I-like receptors (RLRs)-mediated IFN production pathway (Figure 4A). The loss-of-function mutants of OTUD1 seem to attenuate the IRF3 binding and/or deubiquitination activity against IRF3. The authors further identify that the transcription factor forkhead box O3 (FOXO3) plays an essential role in the upregulation of OTUD1 under serum starvation conditions.

Upon *Candida albicans* fungal infection, C-type lectin receptors (CLRs) expressed on macrophages and dendritic cells recognize the fungal cell wall glucans and then activate Src family Tyr kinase and protein kinase Cδ (PKCδ). Subsequently, the PKCδ-mediated phosphorylation of CARD9 induces the CARD9–BCL10–MALT1 (CBM) complex, which activates the NF-κB and MAPK signaling pathways [42]. TRIM62, an E3, reportedly conjugates the K27-linked ubiquitin chain at K125 of CARD9, which promotes CARD9 activation [43]. OTUD1 interacts with CARD9 through the OTU + UIM domain and deubiquitinates CARD9 [44] (Figure 4C). OTUD1 positively regulates CLRs-induced activation of NF-κB and MAPK signaling pathways by potentiating CBM complex formation. Therefore, *Otud1* deficiency impairs cytokine production and antifungal immune responses in vivo. Furthermore, Wu et al. report that *Otud1*-deficient mice show increased susceptibility to dextran sulfate sodium (DSS)-induced colitis, with upregulation of inflammatory cytokines due to the increased NF-κB signaling [45]. On the LPS-Toll-like receptor 4 (TLR4)-mediated NF-κB activation pathway, OTUD1 basically suppresses NF-κB activation by removing the K63-linked ubiquitin chain on K627 of receptor-interacting serine/threonine-protein kinase 1 (RIPK1). This inhibits the recruitment of NEMO, a regulatory subunit of IκB kinase (IKK), which is a crucial kinase for the NF-κB pathway (Figure 4D). The authors also show that the UC-associated G430V mutant of OTUD1 (Figure 4B) fails to deubiquitinate RIPK1. Taken together, these findings indicate that OTUD1 is a suppressor of the NF-κB signaling pathway, and the aberrant OTUD1 activity induces inflammatory bowel diseases such as UC by the hyperactivation of the NF-κB pathway. Thus, OTUD1 seems to be an important suppressor of inflammatory responses through the activation of the NF-κB pathway. Indeed, *Otud1* is 1 of 13 key genes that are crucial for LPS-induced cardiac dysfunction and sevoflurane-induced cardioprotection in mice [46]. Interestingly, OTUD1, as well as the deubiquitinase USP42 and the epigenetic modifier TAF9B/SAGA complex, serves as a hub in osteoarthritis onset and progression [47]. These results suggest that OTUD1 is deeply involved in the development of inflammatory diseases, and highlight its importance as a drug discovery target.

**Figure 4 antioxidants-12-00350-f004:**
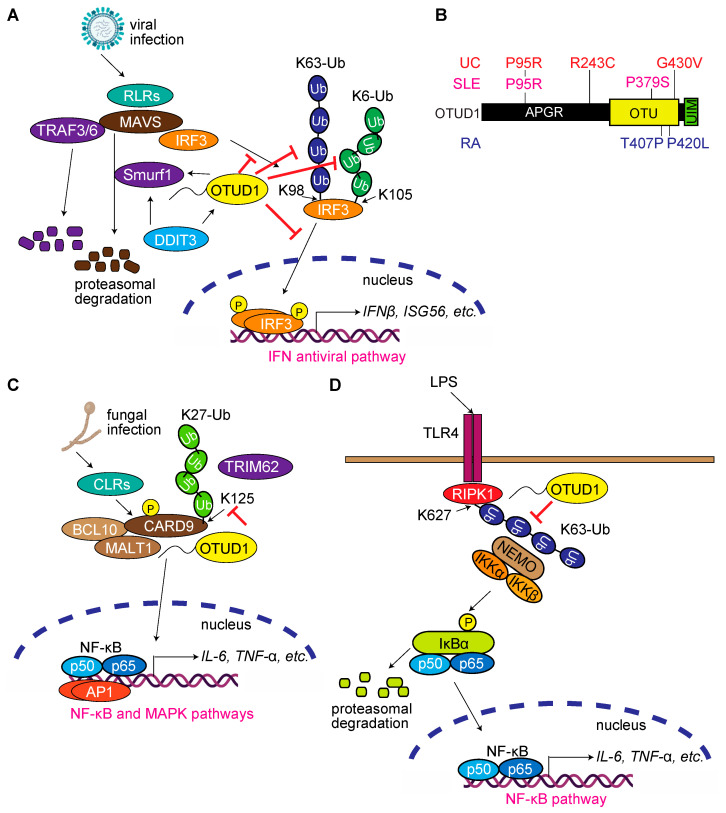
OTUD1 regulates antiviral and innate immune responses, and mutations in *OTUD1* are associated with autoimmune diseases. (**A**) OTUD1 regulates the IRF3-mediated type I IFN production pathway as an antiviral response [37,38,39]. (**B**) Mutations in *OTUD1* are associated with autoimmune diseases [41]. The amino acid mutations in OTUD1 and associated autoimmune diseases are shown. UC, ulcerative colitis; SLE, systemic lupus erythematosus; RA, rheumatoid arthritis. (**C**) OTUD1 regulates CARD9-mediated NF-κB and MAPK signaling pathways upon fungal infection [43,44]. (**D**) OTUD1 cleaves the K63-linked ubiquitin chain on RIPK1 and regulates the NF-κB pathway [45].

### 3.5. OTUD1 Regulates Cell Death Pathways

As described above, OTUD1 binds and regulates the mitochondrial antiapoptotic protein AIF, and this interaction is associated with caspase-dependent apoptosis and caspase-independent parthanatos with OXPHOS-related ROS production (Figure 3C) [36]. Several papers report that OTUD1 is involved in ROS-induced oxidative damage and cell death regulation. Melatonin, secreted by the pineal gland, is involved in circadian rhythms, and has anticarcinogenic, antioxidant, and anti-inflammatory functions. Melatonin upregulates the expression of Bcl-2 interacting mediator of cell death (Bim), a proapoptotic protein [48], as well as DUBs such as OTUD1, USP2, and USP50. However, the *OTUD1* knockdown specifically inhibits melatonin-induced Bim upregulation [49]. Melatonin promotes the OTUD1-mediated stabilization of Bim by the deubiquitination of K3 (Figure 5A). Furthermore, melatonin upregulates Sp1 expression, which directly activates the *OTUD1* promoter. Melatonin markedly reduces tumor growth and size with increased apoptosis of cancer cells, and the expression levels of Bim and OTUD1 are correlated in patients with renal clear carcinoma. Thus, melatonin induces apoptosis by stabilizing Bim through OTUD1 upregulation.

**Figure 5 antioxidants-12-00350-f005:**
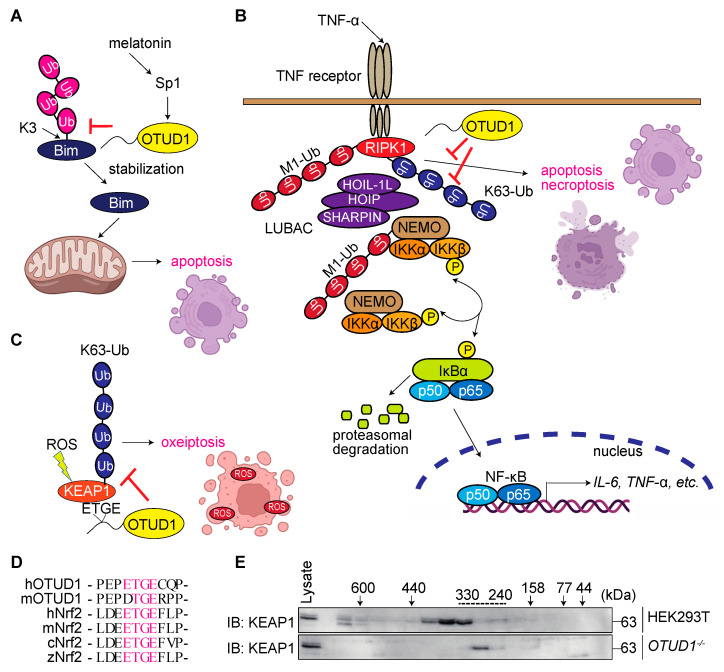
OTUD1 regulates cell death pathways. (**A**) OTUD1 stabilizes Bim and induces apoptosis [49]. (**B**) OTUD1 suppresses the canonical NF-κB and TNF-α-induced apoptosis and necroptosis pathways by cleaving the K63-linked ubiquitin chain [19]. (**C**) OTUD1 binds KEAP1 through the ETGE motif in the APGR of OTUD1. (**D**) The amino acid sequence alignment of ETGE motifs in OTUD1 and Nrf2. h: human, m: mouse, c: chicken, z: zebrafish. (**E**) *OTUD1* deficiency affects the intracellular molecular mass of the KEAP1-containing complex. Cited from Ref. [19].

Linear ubiquitin chain assembly complex (LUBAC) is the only E3 that can generate the N-terminal Met1(M1)-linked linear ubiquitin chain [50]. LUBAC activates the canonical NF-κB signaling pathway and suppresses TNF-α-induced cell death pathways such as apoptosis and necroptosis. We comprehensively screened for DUBs that participate in the regulation of LUBAC-mediated NF-κB activation, and identified OTUD1 as a potent suppressor [19]. In *OTUD1*-deficient cells, the K63-linked ubiquitination of NF-κB signaling factors such as NEMO, RIPK1, IRAK1, and LUBAC subunits was enhanced with the increased NF-κB activation upon TNF-α or IL-1β stimulation (Figure 5B). Thus, OTUD1 is a critical negative regulator of the NF-κB pathway. OTUD1 also suppresses TNF-α-induced apoptosis and necroptosis. Interestingly, the K63-linked ubiquitination of RIPK3, a critical component for necroptosis induction, was upregulated in *OTUD1*-deficient cells, indicating that RIPK3 is an endogenous substrate of OTUD1. Our mass spectrometric analysis reveals that OTUD1 binds KEAP1, a regulator of the transcription factor Nrf2, during ROS-induced oxidative stress [19] (Figure 5C). Importantly, the N-terminal (aa 49–52) APGR of OTUD1 includes an ETGE motif, which is the KEAP-binding site in Nrf2 (Figure 5D). In gel filtration analysis of *OTUD1*-deficient cells, KEAP1 eluted at lower molecular weight fractions as compared to that from parental cells (Figure 5E) with the increased K63-linked ubiquitination of KEAP1. This clearly indicates that KEAP1 and OTUD1 interact at the endogenous level. Moreover, ROS levels were upregulated in H_2_O_2_-treated *Otud1^−/−^* MEFs as compared to parental MEFs, resulting in the enhanced expression of antioxidant genes such as *HO-1* and *Nrf2*, and AIF1-mediated oxidative cell death referred to as oxeiptosis. Therefore, *Otud1^−/−^* mice exhibit susceptibility toward DSS-induced UC-like inflammatory bowel disease, LPS-induced sepsis, and LSP + D-galactosamine-induced acute hepatitis with enhanced cell death and oxidative damage in vivo [19]. These results indicate that OTUD1 is a physiological binding protein and deubiquitinase of KEAP1, which plays an important role in suppressing oxidative-stress-induced oxeiptosis as an antioxidant. The KEAP1-binding of OTUD1 has been independently confirmed [51].

### 3.6. OTUD1 Regulates Acquired Immunity and Hematologic Disorders

Multiple myeloma (MM) is a malignancy of immunoglobulin (Ig)-producing plasma cells and characteristically secretes an abnormal serum monoclonal Ig named M protein. Proteasome inhibitors such as bortezomib, carfilzomib, and ixazomib are reportedly effective in the treatment of MM. Vdovin et al. found that the expression levels of intracellular Ig and OTUD1 are associated with survival after the onset of MM, since OTUD1 regulates Ig production, myeloma cell proliferation, and tumor growth [51]. Furthermore, OTUD1 modulates intracellular Ig-light chain production and sensitivity to proteasome inhibitors in plasma cells. As a cellular mechanism leading to the onset of MM, OTUD1 deubiquitinates peroxiredoxin 4 (PRDX4), an antioxidant protein in the ER, protecting it from ER-associated degradation, and then regulates Ig synthesis (Figure 6A). Although the authors identify that OTUD1 binds KEAP1, a substrate adaptor of the CUL3-dependent E3 complex, the cellular function of the association remains to be elucidated. The inhibition of HSP90 by tanespimycin overcomes proteasome inhibitor resistance in MM with low intracellular Ig. Thus, the OTUD1–PRDX4 axis seems to be a crucial target for proteasome-inhibitor-independent therapy for MM.

Acute graft-versus-host disease (aGVHD) is a serious complication that affects patients who have had a stem cell transplant using cells from a donor, due to the allogeneic response mediated by donor T cells, which induces uncontrolled T cell activation, proliferation, and upregulation of inflammatory cytokines. Recently, Cheng et al. identified OTUD1 as an essential regulator of aGVHD [52]. These authors showed that the mRNA levels of *Otud1*, as well as *Otud4* and *Otud5*, were significantly upregulated in allogeneic T cells, but not syngeneic T cells. Moreover, the authors determine that OTUD1 is abnormally upregulated in CD4^+^ T cells, but not CD8^+^ T cells, of aGVHD patients. OTUD1 levels are positively correlated with the incidence and severity of aGVHD, and, therefore, *Otud1* deficiency suppresses T cell activation and proinflammatory cytokine production, resulting in the alleviation of aGVHD pathogenicity. As the molecular basis, they show that OTUD1 selectively deubiquitinates K1770 on the intermediate Notch-intracellular domain (Notch-ICD) of Notch2, but not Notch1, Notch3, or Notch4, protects it from degradation, and regulates the expression of Notch2 target genes (Figure 6B). The Notch-ICD is essential for OTUD1 to regulate both the T cell receptor signal transduction in activated T cells and the development of aGVHD in mice. Interestingly, a treatment with dapagliflozin, a medicine for type 2 diabetes, increased the ubiquitination of Notch-ICD, suggesting that dapagliflozin may be an inhibitor of OTUD1, since it impaired the production of IFNγ and IL-2 in CD4^+^ T cells, prolonged the survival of aGVHD mice, and attenuated the pathogenesis of the targeted organs. These results indicate that OTUD1 is involved in T-cell-mediated allogeneic responses and that OTUD1 is a therapeutic target for aGVHD.

### 3.7. OTUD1 Regulates Translation and RNA Metabolism

The ubiquitin system plays an important role in ribosome-associated quality control (RQC), and the ZNF598 E3 (Hel2 in yeast)-mediated ubiquitination stalls ribosomes on a polyA stretch through the K63-ubiquitination of uS10 [53,54]. OTUD3 and USP21 were previously identified as negative regulators of ribosome stalling [55]. Recently, Snaurova et al. reported that OTUD1 interacted with elongating ribosomes, and suppressed polyA-induced ribosome stalling by forming polysomes (Figure 6C) [56]. Although detailed examinations in vivo are required, OTUD1 seems to antagonize ZNF598 and suppress the stalling pathway, which facilitates rare-codon-mediated decay.

### 3.8. OTUD1 in Vascular Remodeling

Angiotensin II (Ang II), a peptide hormone involved in vasocontraction and increasing blood pressure, also affects hypertension and hypertensive vascular remodeling. Huang et al. recently showed that among OTU family DUBs, only the *OTUD1* mRNA is upregulated in the aortic endothelium of Ang II-treated mice, and vascular remodeling and collagen deposition are attenuated in Ang II-treated *Otud1*-deficient mice [57]. Furthermore, the authors identify that OTUD1 exacerbates the Ang II-induced endothelial-to-mesenchymal transition. FLAG–OTUD1-pulldown assays followed by mass spectrometric analyses of human umbilical vein endothelial cells suggest that SMAD3 binds OTUD1. Indeed, OTUD1 associates with the C-terminal MH2 domain of SMAD3 and deubiquitinates K48- and K63 ubiquitin chains from SMAD3, thus, contributing to SMAD3 activity, SMAD3/SMAD4 complex formation, and nuclear translocation (Figure 6D). OTUD1 overexpression aggravates vascular remodeling and collagen deposition by regulating SMAD3 in mice. Thus, OTUD1 is a potential therapeutic target for diseases related to vascular remodeling by its interactions with SMAD3.

**Figure 6 antioxidants-12-00350-f006:**
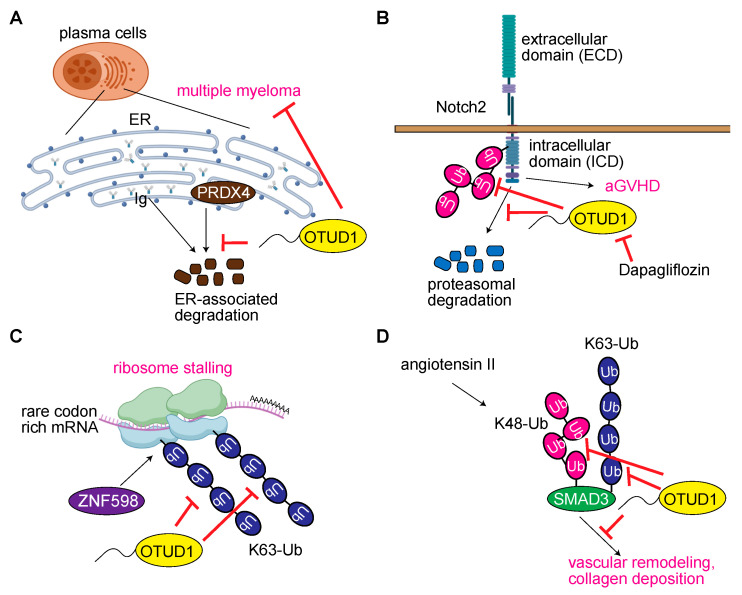
Extended cellular functions of OTUD1. (**A**) OTUD1 affects multiple myeloma through the regulation of ER-associated degradation of PRDX4 [51]. (**B**) OTUD1 is involved in aGVHD through the regulation of Notch2-ICD degradation, and dapagliflozin may be an inhibitor of OTUD1 [52]. (**C**) OTUD1 resolves stalled translation [56]. (**D**) OTUD1 promotes angiotensin-II-induced vascular remodeling by deubiquitinating SMAD3 [57].

### 3.9. OTUD1 Is Involved in Stress Responses in a Wide Range of Species

OTUD1 is encoded by a gene that is characteristically induced by the rearing nutritional status and stress of animals such as pigs, chickens, and fish. Thus, OTUD1 was identified as one of the upregulated genes in muscle tissues from 20% rapeseed meal (RSM)-treated Norwegian Landrace growing–finishing pigs, as compared with the soybean-meal-fed pigs [58]. This comparison revealed that in RSM-fed pigs, 63 genes were downregulated whereas 57 genes were upregulated. The gene ontology analysis suggests that genes encoding proteins are involved in p53-mediated signaling, negative growth regulation, and protein synthesis is downregulated in RSM-fed pigs, although those of metabolic controllers, proteolysis, lipolysis, fatty acid oxidation, and oxidative stress are upregulated. Since OTUD1 is involved in p53 and oxidative stress regulation, these biological responses may be inevitable. Similarly, pigs fed a diet supplemented with hydroxy-methionine expressed OTUD1, MURF1, and so on in the longissimus lumborum muscle [59]. Since dietary methionine affects protein metabolism, growth performance, and oxidative stress, the expression of OTUD1 may influence these cellular functions.

In the muscle tissue of high-yielding broiler line chickens, several muscle myopathies such as woody breast myopathy were identified, and a high level of OTUD1 expression was detected in the musculoskeletal system at day 21 [60]. Although the authors suggest the potentially higher degradation of muscle protein, the details remain elusive.

When zebrafish larvae were exposed to lethal cold stress (10 °C) and rewarmed to normal temperature (28 °C), *otud1,* as well as *trib3* and *dusp5,* was identified as a hub gene of cold-induced damage [61]. The gene ontology analysis suggests that the autophagy, FoxO, and MAPK pathways are elevated to survive against cold stress, whereas the apoptosis, necroptosis, and p53 signaling pathways seem to be responsible for cold-induced mortality. Therefore, OTUD1 may have a crucial function in cold-stress resistance by regulating cell death and signaling pathways in zebrafish. Collectively, these results indicate that OTUD1 plays an important role as an environmental stress response factor in a wide range of animal species.

### 3.10. Miscellaneous Functions of OTUD1

A genome-wide association study (GWAS) by Cozier et al. identified two SNPs, rs1398024 (G>T) and rs11013452 (G>A, T), on the *C10ORF67–OTUD1* intergenic region of chromosome 10p12, that are strongly associated with sarcoidosis in African-American women [62]. Sarcoidosis is characterized by the abnormal growth of inflammatory cells (granulomas) in the lung, lymph nodes, skin, eyes, and other organs. Although the details remain obscure, sarcoidosis is triggered by infection or chemicals, as well as immune responses, and, therefore, the impaired regulation of OTUD1 may be related to sarcoidosis development. Future analyses should yield interesting findings.

## 4. Conclusions

In this review, we comprehensively summarized the functions of OTUD1, a DUB that has recently received keen attention. OTUD1 exhibits strong activity against the K63 ubiquitin chain in vitro, but also reportedly cleaves K6, K27, K33, K48, and K63 ubiquitin chains in cells. These activities reveal that OTUD1 is involved in the regulation of numerous cellular functions, such as oncogenic protein regulation, antiviral signaling, inflammation, innate and acquired immunity, cell death, and ribosomal function. In addition, there are OTUD1-related diseases, such as various cancers, autoimmune diseases, sepsis, inflammatory bowel disease, aGVHD, arteriosclerosis, and so on. It is highly likely that ROS production and oxidative damage are involved in the regulation of these cell functions and diseases related to OTUD1, and the KEAP1–OTUD1 axis plays an important role. The development of OTUD1-targeted inhibitors, activators, and KEAP1-binding inhibitors is expected to provide new therapeutic drugs. Dapagliflozin, an SGLT-2 inhibitor for type 2 diabetes treatment, reportedly has the potential to inhibit OTUD1, and its binding model has also been proposed [52]. Detailed analyses in vitro and in vivo will be important, and new derivatives based on dapagliflozin may be developed. OTUD1 will become even more important in the future as a therapeutic target with links to cell functions and diseases.

## Figures and Tables

**Figure 1 antioxidants-12-00350-f001:**
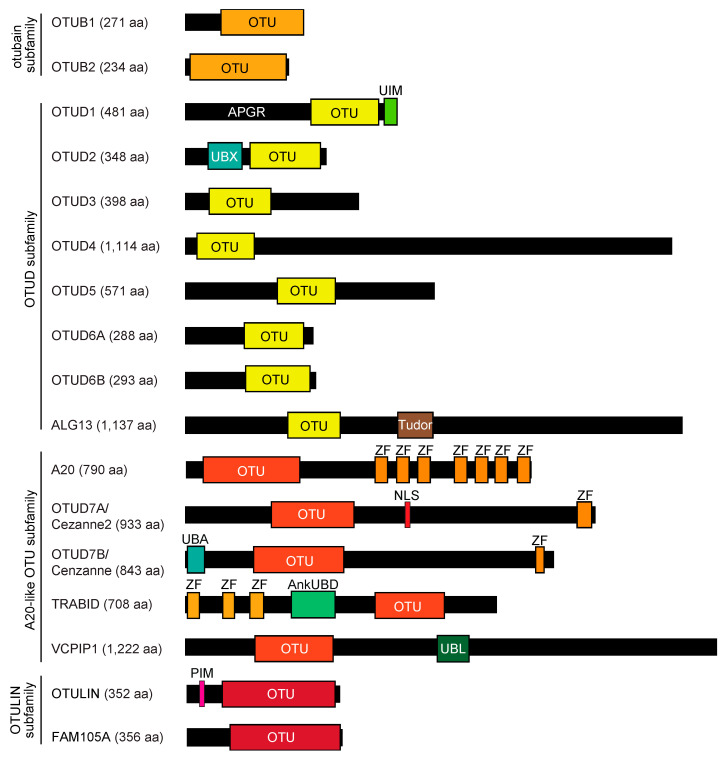
Domain structures of OTU family deubiquitinating enzymes (DUBs). The seventeen human OTU family DUBs are classified into four subfamilies, and the total numbers of amino acid residues and domain organizations are shown. OTU, ovarian tumor protease; APGR, Ala-, Pro-, and Gly-rich region; UIM, ubiquitin-interacting motif; UBX, ubiquitin regulatory X; Tudor, Tudor domain; ZF, zinc finger; NLS, nuclear localization signal; UBA, ubiquitin-associated; AnkUBD, ankyrin repeat ubiquitin-binding domain; UBL, ubiquitin-like; PIM, PUB domain-interacting motif.

## Data Availability

No new data were created or analyzed in this study. Data sharing is not applicable to this article.

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
