# Peer review of "Pleiotropic Roles of a KEAP1-Associated Deubiquitinase, OTUD1"

_antioxidants, 2023, doi:10.3390/antiox12020350_

Round 1

Reviewer 1 Report

The article has an important topic for the medical field. 

I would like also to congratulate the authors for their hard work. I have only one remark.

Please wrote the full name (were it it possible) for all abbreviations, such AKT, PTEN, TGF

Author Response

We thank you for the reviewer’s comment. The abstract has a limit of 200 words, so it cannot be included in the abstract. However, we have added their full names of AKT, PTEN, and TGF to lines 157, 156, and 141, respectively, which appear for the first time in the revised manuscript. Furthermore, we showed the full names of KEAP1 (line 70), SMAD (line 141), and Nrf2 (lines 252-253).

Reviewer 2 Report

The review article submitted by Oikawa et al. entitled "Pleiotropic roles of a KEAP1-associated deubiquitinase, OTUD1" to the SI 10th Anniversary of Antioxidants—Review Collection deeply reviews the knowledge about a an ovarian tumor protease (OTU) family DUB called OTUD1. Briefly, the authors wrote about several aspects such as:

The Pleiotropic cellular functions of OTUD1; the OTUD1 functions as a tumor regulator; OTUD1 regulates Yes-associated protein (YAP) signaling; OTUD1 regulates the degradation of myeloid cell leukemia 1 (MCL1); OTUD1 regulates antiviral and innate immune responses, and its dysfunction is associated with autoimmune and inflammatory diseases; how OTUD1 regulates cell death pathways; how OTUD1 regulates acquired immunity and hematologic disorders; mechanisms of how OTUD1 regulates translation and RNA metabolism; the role of OTUD1 in vascular remodeling; how OTUD1 is involved in stress responses in a wide range of species; and the miscellaneous functions of OTUD1.

Thus, in summary, in the opinion of this reviewer, this manuscript is very well-written and structured. The manuscript comprehensively summarized the functions of OTUD1, a DUB that has recently received keen attention. OTUD1 exhibits strong activity against the K63 ubiquitin chain in vitro, but also reportedly cleaves K6, K27, K33, K48, and K63 ubiquitin chains in cells, uncovering that OTUD1 is involved in the regulation of numerous cellular functions, such as oncogenic protein regulation, antiviral signaling, inflammation, innate and acquired immunity, cell death, and ribosomal function.

However, the authors should correct two minor points before the acceptance of the manuscript:

In line 69 an additional reference reporting the fact that virus can encode a Ubiquitin-related enzyme should be added (e.g. https://www.nature.com/articles/s41598-018-21872-2). Additionally, the software used to build up the figures should be acknowledged.

Author Response

We thank you for the reviewer’s evaluation and for pointing out the very important paper. We have quoted the paper as a new reference 12 on line 78. We added acknowledgments to the software on line 587 based on the reviewer's instructions.

The corresponding author is a premium member of BioRender and is permitted to use BioRender icons in publications. In addition, our university is an official user of Adobe Creative Cloud.

Reviewer 3 Report

In the submitted review manuscript Oikawa et al. gave a very extensive overview of versatile roles of OTU deubiquitinase 1 (OTUD1) enzyme. This manuscript is very well written, comprehensive and very informative. It is well structured and easy to read, devoid of all unnecessary sentences. Graphics are clear and informative.

There are just few minor suggestions:

1) For the sake of non-ambiguity, full names of amino acids should be written before introduction of their one- or three-letter abbreviations.

2) Generic drug names should not be capitalized. The same is with amino acids three-letter abbreviations.

3) Statement in line 63 "...it will be an important therapeutic target." is little bit too bold, almost like precognition.

4) Abbreviation "DUB" should be explained in Figure 1 legend.

5) Line 193: Should be "TEAD family of transcription factors".

6) Line 198: I suppose you meant TNBC cell line MDA-MB-231.

7) Lines 291 and 337: Since you wrote about amino acid mutations, protein symbol OTUD1 should be provided, not gene symbol OTUD1.

8) Since Figure 5E is the Figure 5C from https://doi.org/10.1038/s41419-022-05145-5 I hope authors have acquired rights for reproduction.

Author Response

We thank you for the reviewer’s evaluation and helpful comments.

1) For the sake of non-ambiguity, full names of amino acids should be written before introduction of their one- or three-letter abbreviations.

The full names of amino acids are shown before the one- or three-letter abbreviations on the first page of the revised manuscript.

2) Generic drug names should not be capitalized. The same is with amino acids three-letter abbreviations.

We have corrected the capitalization of the drugs as indicated. We believe that it is common in biochemistry to write the three-letter abbreviation for amino acids in capital letters, so let's leave these as they are.

3) Statement in line 63 "...it will be an important therapeutic target." is little bit too bold, almost likeprecognition.

According to the reviewer’s comment, we removed the sentence on line 72.

4) Abbreviation "DUB" should be explained in Figure 1 legend.

We explained it to “deubiquitinating enzymes (DUBs)” on line 121.

5) Line 193: Should be "TEAD family of transcription factors".

We corrected it as indicated on line 242.

6) Line 198: I suppose you meant TNBC cell line MDA-MB-231.

Thank you for pointing out our mistake. We corrected it as indicated on line 247.

7) Lines 291 and 337: Since you wrote about amino acid mutations, protein symbol OTUD1 should be provided, not gene symbol OTUD1.

As indicated, we fixed the plain typeface OTUD1 instead of italics on lines 345 and 391.

8) Since Figure 5E is the Figure 5C from https://doi.org/10.1038/s41419-022-05145-5 I hope authors have acquired rights for reproduction.

We thank you for the reviewer’s comment. As noted in the following link (https://s100.copyright.com/AppDispatchServlet?title=OTUD1%20deubiquitinase%20regulates%20NF-κB-%20and%20KEAP1-mediated%20inflammatory%20responses%20and%20reactive%20oxygen%20species-associated%20cell%20death%20pathways&author=Daisuke%20Oikawa%20et%20al&contentID=10.1038%2Fs41419-022-05145-5&copyright=The%20Author%28s%29&publication=2041-4889&publicationDate=2022-08-08&publisherName=SpringerNature&orderBeanReset=true&oa=CC%20BY), our Cell Death Dis paper is open access and CCBY licensed. Therefore, it does not require permission for reuse if proper citation is given. Although we cited the reference, we added “Cited from Ref [19]” on line 418 for clarity.